# Effects of Citric Acid and Humic-like Substances on Yield, Enzyme Activities, and Expression of Genes Involved in Iron Uptake in Tomato Plants

Fabián Pérez-Labrada [1], Adalberto Benavides-Mendoza [2], Antonio Juárez-Maldonado [1], Susana Solís-Gaona [3] and Susana González-Morales [4,*]

1   Departamento de Botánica, Universidad Autónoma Agraria Antonio Narro, Saltillo 25315, Mexico; fabian.perezl@uaaan.edu.mx (F.P.-L.); juma841025@gmail.com (A.J.-M.)
2   Departamento de Horticultura, Universidad Autónoma Agraria Antonio Narro, Saltillo 25315, Mexico; abenmen@gmail.com
3   United Phosphorus Limited, UPL, Saltillo 25290, Mexico; susana.solis@upl-ltd.com
4   Departamento de Horticultura, CONAHCyT-Universidad Autónoma Agraria Antonio Narro, Saltillo 25315, Mexico
*   Correspondence: sgonzalezmo@conahcyt.mx or qfb_sgm@hotmail.com Tel.: +52-844-1222471

**Abstract:** Iron (Fe) deficiency is a common abiotic stress on plants growing in calcareous soils where low organic matter content, high carbonate–bicarbonate concentration, and high pH precipitate Fe in unavailable forms. Enzymatic activity is a mechanism for plants to access soil nutrients; enzymes such as $H^+$-ATPase, phosphoenolpyruvate carboxylase (PEPC), and the intracellular enzyme ferric reduction oxidase (FRO) are involved in Fe absorption. The effects of the application of citric acid (CA) and humic-like substances (HLS) on the yield, $H^+$-ATPase, PEPC, and FRO enzyme activity, and expression of *LeHA1*, *LePEPC1*, and *LeFRO1* genes in tomato plants grown under calcareous soil were studied. CA and HLS improved the SPAD units and increased the number of harvested fruits and yield per plant. Temporary alterations in enzyme activity, which reduced PEPC and FRO activity in roots, were documented. In leaf tissue, CA resulted in lower expression of *LeHA1* and *LePEPC1* and the induction of *LeFRO1* expression, whereas HLS application resulted in higher expression of *LePEPC1* and *LeFRO1*. In roots, *LeHA1* expression increased with HLS, whereas *LePEPC1* and *LeFRO1* showed lower expression with CA and HLS, respectively. The application of CA and HLS through a nutrient solution in combination with Fe-chelate can improve Fe nutrition in tomato plants potted in calcareous soil by inducing temporal alterations in PEPC and FRO enzyme activity and *LeFRO1* and *LeHA1* gene expression.

**Keywords:** ferric chelate reductase; $H^+$-ATPase; humic acid; phosphoenolpyruvate carboxylase

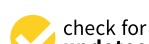



## 1. Introduction

Iron (Fe) plays a crucial role in photosynthesis and mitochondrial respiration, and, as a cofactor of enzymes such as superoxide dismutase, acotinase, lipoxygenases, nitrogenase, carotenoid cleavage oxygenases/dioxygenase, lutein cleavage dioxygenase, and cupins, among others, it is generally found in high proportions in chloroplasts—up to 80% [1]. Therefore, plants require an average of $\sim 10^{-9}$–$10^{-4}$ M Fe ions for optimal development [2]. Fe is taken up from the soil; however, despite its abundance in soil, a high percentage of Fe exists as $Fe^{3+}$, which is highly insoluble and has very low availability [3]. This subsequently generates abiotic stress in the form of Fe deficiency. An Fe-deficient environment is generated in calcareous soils, where Fe precipitates in forms unavailable to plants due to high pH (>7.5), high content of carbonates ($CaCO_3$), and low content of organic matter [4]. Similarly, the high bicarbonate ($HCO_3^-$) concentration (in conjunction with the high pH) decreases Fe mobility within the plant, triggering iron deficiency chlorosis (IDC), which is



characterized by the loss of chlorophyll, interveinal chlorosis of young leaves, deformation of leaves, reduction of photosynthesis and growth, and decrease in plant production [1].

Because most of the Fe in the soil is present as $Fe^{3+}$, plants use enzymatic mechanisms to access it, employing a reduction-based strategy (I) or a chelation-based strategy (II) [5]. Strategy II plants (with *Oryza sativa* as a model) secrete phytosiderophores using transporter of mugineic acid 1 (TOM1), which chelates $Fe^{3+}$ [6]. The phytosiderophores-$Fe^{3+}$ complex is subsequently absorbed by yellow stripe 1 (YS1) or yellow stripe-like 1 (YSL) [7]; $Fe^{2+}$ uptake by iron-regulated transporter (IRT) 1/2 may also occur [8].

In contrast, strategy I plants (with *Arabidopsis thaliana* L. as a model) initially induce rhizospheric acidification through a membrane-associated proton pump, $H^+$-ATPase [9], which releases $Fe^{3+}$ from the soil, and through ferric reduction oxidase (FRO), which reduces $Fe^{3+}$ to $Fe^{2+}$. $Fe^{2+}$ is captured and finally absorbed and transported into the plant by an iron-regulated transporter-like protein (ZIP) family transporter, IRT1, and natural resistance-associated macrophage protein (NRAMP1) [6,10]. The *FRO*, *IRT1*, and *NRAMP1* genes are regulated transcriptionally by the central basic helix–loop–helix (bHLH) FER-like Fe deficiency-induced transcription factor (FIT) [11]. Similarly, phenolic compounds such as coumarins [12,13] and organic acids [6] are exuded by roots and are involved in the Fe-making machinery by facilitating the solubilization and reduction of unavailable Fe in the soil.

Under conditions of Fe deprivation, plants regulate the enzymatic activity associated with Fe access at the genic level. For example, strategy I plants, such as tomato (*Solanum lycopersicum* L.), substantially increase FRO enzymatic activity in the roots [14]. In pea plants (*Pisum sativum* L.), in addition to higher FRO activity, overexpression of *FRO1*, *IRT1*, and *HA1* was observed in the roots [15]. Similarly, in cucumber (*Cucumis sativus* L.) an increase in *CsHA1* transcripts was reported in a higher proportion in the roots than in the leaves [16], whereas overexpression of *HA6*, which allowed for the outflow of $H^+$ from the root, was reported in a citrus species [17]. A key regulator in the activation of these mechanisms is FIT, which facilitates Fe uptake [18].

The application of synthetic chelates is commonly recommended for the prevention and reduction of Fe deficiencies in plants grown in calcareous soils. However, these products can be lost by leaching or adsorbing on the soil particles because of the high pH values and high carbonate content of the soil and because of their high affinity with calcium or magnesium [19]. In this sense, citric acid (2-hydroxy-propane-1,2,3-tricarboxylic acid) and leonardite-derived substances (humic, fulvic acid, and humin) improve Fe nutrition in calcareous soils by increasing Fe availability and optimizing synthetic chelate efficiency [20,21].

The exogenous application of citric acid in nutrient solution was found to increase leaf tissue Fe content in tomatoes potted in calcareous soils [21]. Citric acid improves Fe bioavailability by converting Fe to its plant-available form [22]. Similarly, the application of leonardite plus ferrous sulfate heptahydrate was found to improve the fertility of calcareous vineyard soils by increasing their Fe content [23]. The application of Fe complexed with extractable humic substances in water and Fe-citrate led to the overexpression of *FRO*, *IRT1*, and *NRAMP* in cucumber [24] as well as a modulation of the transcripts of the gene encoding $H^+$-ATPase in tomato [25]. Moreover, this strategy also provides a long-term supply of Fe to plants grown in calcareous soils [26]. The aim of this study was to determine the yield, $H^+$-ATPase, phosphoenolpyruvate carboxylase (PEPC), and FRO enzyme activity, and expression of *LeHA1*, *LePEPC1*, and *LeFRO1* in tomato plants potted in calcareous soil under the application of citric acid (CA) and humic-like substances (HLS).

## 2. Materials and Methods

### 2.1. Plant Growth Conditions

Tomato (*Solanum lycopersicum* L. cv. 'Río Grande') (Crow Seed) plants were grown in a greenhouse with a polyethylene cover and 70% natural irradiance (with recorded averages of 50–60% relative humidity, 32 °C ambient temperature, 736.5 µM m$^{-2}$ s$^{-1}$ pho-

tosynthetically active radiation, and 430 ppm $CO_2$ concentration). The tomato seeds were grown for 35 days in a tray containing a germination substrate, and then the seedlings were transplanted into pots containing 9 L of calcareous soil (pH 8.5, 0.2% organic matter, and 5 mg $kg^{-1}$ Fe content). The plants were fertilized from transplantation to the end of the experiment with Steiner solution [27] at 100% concentration, which contained 4.5 mmol $Ca(NO_3)_2 \cdot 4H_2O$, 2.0 mmol $MgSO_4$, 0.7 mmol $KNO_3$, 2.0 mmol $K_2SO_4$, and 1.6 mmol $KH_2PO_4$. Micronutrients were applied from a stock solution containing $H_3BO_3$ (2.86 g $L^{-1}$), $MnSO_4 \cdot H_2O$ (2.15 g $L^{-1}$), $ZnSO_4 \cdot 7H_2O$ (0.39 g $L^{-1}$), and $CuSO_4 \cdot 5H_2O$ (0.078 g $L^{-1}$), and $Na_2MoO_4 \cdot 5H_2O$ (0.09 mg $L^{-1}$). Finally, Fe was applied in the form of Fe gluconate-ethylenediaminetetraacetic acid (EDTA)-type chelate (at 3 mg $L^{-1}$), except in the Fe-deprivation treatment, where it was not added. The pH of the nutrient solution was maintained at 6.3 with an electrical conductivity of 2.0 dS $cm^{-1}$.

The treatments studied were CA (0.1 mM + Fe gluconate-EDTA), HLS (400 µL $L^{-1}$ + Fe gluconate-EDTA), "without organic amendment" (WOA, only Fe gluconate-EDTA), and an Fe-deprivation control (ID, no organic acid and Fe in solution). The concentrations used were selected according to previous results cited by Pérez-Labrada et al. [21]. The citric acid used was food-grade (99.9% purity). The HLS (Arysta LifeScience) contained 10 g $L^{-1}$ humic acid carbon, 90 g $L^{-1}$ fulvic acid carbon, 6 g $L^{-1}$ total nitrogen, 6 g $L^{-1}$ urea nitrogen, and 38 g $L^{-1}$ water soluble $K_2O$ with a density of 1.15 g $mL^{-1}$ and at pH 9.2. The treatments CA and WOA were applied daily through localized irrigation during plant development. The HLS treatment was only applied weekly with the corresponding irrigation volume (15 total applications during crop development). The control (ID) was irrigated daily through localized irrigation during plant development. All treatment compositions were added to the nutrient solution. The nutrient solution was provided to the plants through localized irrigation from transplanting until the end of the experiment.

Considering the phenology of the tomato plant, the sampling dates of leaf-root tissue were chosen according to these stages: vegetative growth (27 days after transplantation, DAT), flowering-anthesis (49 DAT), and physiological maturity of fruits of the 2nd bunch of fruit (89 DAT). We collected the three youngest leaves that had completely expanded, placed them in aluminum bags, immediately froze them with liquid $N_2$, and stored them at –80 °C. Root tissue was collected on the same leaf-sampling date; 5 cm of secondary roots (obtained at 10 cm depth and 5 cm stem distance) were collected, briefly washed with deionized water, placed in aluminum bags, immediately frozen with liquid $N_2$, and stored at –80 °C. Plant growth parameters were determined at 27, 49, and 84 DAT. The stem diameter was measured with a sliding caliper 150 mm between the first and second pair of true leaves from the base. Plant length was measured with a flexometer from the base of the stem (soil surface) to the distal growing apex. On fully expanded young leaves, SPAD-unit readings were taken using a Minolta SPAD-502 chlorophyll meter (Konica Minolta, Inc., Osaka, Japan). SPAD values were taken as an average of three readings taken at different locations from the base to the apex of each leaf. Finally, the yield per plant was calculated as the sum of the total number of harvested fruits.

### 2.2. Enzymatic Activity

Briefly, the enzyme extract was obtained by homogenizing the tissue in liquid $N_2$ and adding buffer at 4 °C (50 mM Tris-HCL, pH 7.5, 10% glycerol, 20% PVPP, 10 mM $MgCl_2$, 1 mM EDTA, 14 mM β-mercaptoethanol, 1 mM PMSF, and 10 µg $mL^{-1}$ leupeptin; Sigma-Aldrich, St. Louis, MO, USA). The extract was then filtered through a nylon micropore (0.45 µm pore size) and centrifuged at 10,000 rpm for 15 min. The supernatant was centrifuged again at 10,000 rpm for 30 min, and the precipitate was resuspended in the same buffer and stored at –80 °C [28] until use in the determination of $H^+$-ATPase and PEPC activity. $H^+$-ATPase activity (EC 7.1.2.1) was determined using a spectrophotometric method [29], coupling the hydrolysis of ATP to the oxidation of NADH [28]; 100 µL of extract was taken, and 25 mM MOPS-BTP buffer (pH 6.5) was added, containing 250 mM sucrose, 50 mM KCl, 1 mM ATP, 1 mM PEP, 0. 25 mM NADH, 15 µg $mL^{-1}$ lactate de-

hydrogenase (EC 1.1.1.27), 30 µg mL$^{-1}$ pyruvate kinase (EC 2.7.1.40), and 0.015% Brij$^{®}$ 58 (Sigma-Aldrich). Changes in absorbance were measured in a spectrophotometer at 340 nm. The H$^+$-ATPase activity was calculated using the NADH standard curve. Phosphoenolpyruvate carboxylase activity (PEPC, EC 4.1.1.31) was determined by coupling its activity to the malate dehydrogenase catalyzed by the oxidation of NADH [30]; to initiate the reaction, 100 µL of extract was taken, and standard buffer containing Tris-HCl (100 mM, pH 8.0), MgCl$_2$ (5 mM), PEP (2.5 mM), NADH (0.2 mM), NaHCO$_3$ (10 mM), and MDH (15 µg mL$^{-1}$) was added. Changes in absorbance were quantified at 340 nm. PEPC activity was calculated using the NADH standard curve.

Chelate ferric reductase activity (FRO, EC 1.16.1.7) was quantified spectrophotometrically by Fe (II)-BPDS concentration according to Romera et al. [31]; 20 mg of tissue was placed in an Eppendorf tube, and 2 mL of CaSO$_4$-7H$_2$O (0.2 mM) was added and allowed to stand for 5 min. Then, the sample was centrifuged (5 min at 10,000 rpm at 4 °C), and the precipitate was recovered and placed in a new tube, where 10 mL of fresh nutrient solution (without Fe) supplemented with 0.3 mM BPDS (Sigma-Aldrich) and 100 µM Fe(III)-EDTA (Sigma-Aldrich) was added. The pH of the solution was adjusted to 5.5 with 5 mM MES-NaOH (Sigma-Aldrich). It was incubated for 1 h in the dark at room temperature, and finally, the absorbance was measured at 535 nm. BPDS forms a red water-soluble complex with Fe$^{2+}$ and only a weak complex with Fe$^{3+}$. The amount of reduced iron was calculated by the concentration of the Fe$^{2+}$-BPDS complex using an extinction coefficient of 22.14 mM cm$^{-1}$. The protein content of the microsomal fraction membranes was determined using 5 µL of the enzyme extract and 250 µL of Bradford reagent, which were incubated at room temperature for 5 min. Afterward, the absorbance was measured at 630 nm in an ELISA plate reader (BioTek, ELx808 model, Winooski, VT, USA) with BSA as a protein standard [32].

### 2.3. Real-Time Reverse-Transcriptase PCR

The RNA of leaves and roots was extracted using TRIzol reagent [33]; the tissue was ground in liquid N$_2$, 100 mg was taken and placed in an Eppendorf tube, and immediately after, 1 mL of TRI Reagent$^{®}$ (MRC, TR 118) was added; the mixture was homogenized gently and incubated for 5 min at room temperature. Subsequently, 200 µL of chloroform was added, shaken vigorously, and incubated for 15 min. Then, the samples were centrifuged at 12,000 rpm (15 min at 4 °C), and the supernatant was recovered and placed in a new Eppendorf tube. Then, 500 µL of isopropanol (4 °C) was added, mixed gently, and incubated for 10 min at room temperature. The sample was centrifuged for 10 min at 12,000 rpm and 4 °C, and the supernatant was removed by removing excess isopropanol from the formed RNA pellet. The RNA pellet was washed with 500 µL of 70% ethanol (4 °C), and excess was removed and allowed to dry. The pellet was suspended in 50 µL of water (dissolved at 60 °C). Finally, the RNA solution was treated with DNase I (Sigma Aldrich) and stored at 4 °C. The RNA quantity and quality were determined using a UV-Vis spectrophotometer (260/280 nm ratio) and via denaturing electrophoresis, respectively. An ImProm-II$^{TM}$ Reverse Transcription System Kit (Promega, Madison, WI, USA) was used for the synthesis of cDNA following the manufacturer's instructions; cDNA was synthesized from 1 µg of RNA sample. The primers corresponded to the endogenous internal control gene (*ACT*) and study genes *LeFRO1*, *LePEPC1*, and *LeHA1* (Table 1). For *LeFRO1* and *LePEPC1*, the sequences cited by Paolacci et al. [34] and Diamantopoulous et al. [35], respectively, were considered. The primers were designed using AmplifiX 17.0 (CNRS by Nicolas Jullien, Marseille, France), OligoAnalyzer 3.1 (Integrated DNA Technologies IDT, Coralville, IA, USA), and Primer-BLAST (National Center for Biotechnology Information, Bethesda, MD, USA).

**Table 1.** Sequences of primers used for gene analysis.

| Name Gene | Nomenclature | Forward Primer 5'-3' | Reverse Primer 5'-3' | Tm (°C) |
|---|---|---|---|---|
| Actin | *ACTIN* | CCCAGGCACACAGGTGTTAT | CAGGAGCAACTCGAAGCTCA | 60 |
| H$^+$-ATPase | *LeHA1* | GAACCCTTCATGGGCTCCAA | GCAACTCACGTAGCCTAGCA | 60 |
| PEPC | *LePEPC1* | TGCTGCATTGTTCGACAAGC | CAAAAGTTCGCCGAAAGACAAC | 60 |
| FRO | *LeFRO1* | GCGGTGTTGAATATGCTAATC | AAACTTTCCATCTCCCTATCG | 60 |

Real-time PCR was performed in a final volume of 20 µL. For *ACT*, 10 µL of SYBR® Select Master Mix (Applied Biosystems, Foster City, CA, USA), 0.10 µL of forward primer (72 nM), 0.08 µL of reverse primer (60 nM), 2 µL of cDNA, and 7.82 µL of nuclease-free water were added. For *LeHA1*, 10 µL of SYBR® Select Master Mix (Applied Biosystems), 0.13 µL of forward primer (100 nM), 0.13 µL of reverse primer (100 nM), 2 µL of cDNA diluted 1:5, and 7.74 µL of nuclease-free water were added. For *LePEPC1*, 10 µL of SYBR® Select Master Mix (Applied Biosystems), 0.13 µL of forward primer (100 nM), 0.27 µL of reverse primer (200 nM), 2 µL of cDNA, and 7.60 µL of nuclease-free water were added. For *LeFRO1*, 10 µL of SYBR® Select Master Mix (Applied Biosystems), 0.13 µL of forward primer (100 nM), 0.13 µL of reverse primer (100 nM), 2 µL of cDNA, and 7.74 µL of nuclease-free water were added. The *ACT* gene was used to normalize the expression ratio of each gene, and changes in expression were calculated using the standard relative curve method [36].

*2.4. Data Analysis*

A completely randomized design was used for the experimental development, with 15 biological replicates per treatment. The experimental unit was an individual plant in a pot. One-way analysis of variance (ANOVA) and Fisher's least significant difference test ($p < 0.05$) were applied to the plant growth parameters. Data of enzymatic activity were analyzed using a two-way ANOVA and a multiple means comparison using Fisher's least significant difference test ($p < 0.05$), in which one factor was the organ evaluated (root and leaf), and the other factor was the treatment applied. The gene expression was normalized compared to the internal reference gene (*ACT*). A one-way Kruskal–Wallis nonparametric test by ranks was applied to the gene expression data ($p < 0.05$). All statistical analyses were performed using IBM SPSS v. 19.0.

**3. Results**

*3.1. Plant Growth*

The application of CA and HLS resulted in statistically significant differences ($p < 0.05$) in the growth parameters evaluated (Table 2) compared to the control (ID). For example, the addition of CA to the fertilizer solution increased stem diameter (14% at 84 DAT), plant height (39% at 27 DAT), and leaf number (43% at 49 DAT). The weekly application of HLS increased SPAD units by 110% and 223% at 49 and 84 DAT, respectively, compared to ID. Similarly, the total fruit harvested and maximum production increased by 93% and 265%, respectively, in HLS-treated plants. In addition, the use of AC and HLS stimulated stem diameter at different sampling times up to 15% with respect to WOA. Plant height was lower under HLS at 27 and 49 DAT compared to CA and WOA. Plants treated with HLS and CA showed an improvement in leaf number at 27 and 84 DAT compared to WOA. The CA and WOA treatments increased this variable by 18% over HLS at 49 DAT. SPAD units were improved with CA and HLS with respect to WOA by up to 14.8%. Finally, HLS showed a higher number of fruits harvested and a higher yield per plant than the WOA and CA treatments.

**Table 2.** Plant growth parameters of tomato plants treated with CA and HLS.

| Sampling | Treatment | Stem Diameter (mm) | Plant Height (cm) | Number of Leaves | SPAD-Unit | Total of Fruit Harvest | Production per Plant (kg) |
|---|---|---|---|---|---|---|---|
| 27 DAT | CA | 10.80 ± 0.52 a[†] | 45.70 ± 3.38 a | 13.20 ± 0.45 a | 54.19 ± 2.78 a | - | - |
| | HLS | 10.24 ± 0.96 a | 39.40 ± 4.39 b | 13.40 ± 0.89 a | 53.65 ± 2.38 a | - | - |
| | WOA | 10.00 ± 0.29 a | 45.50 ± 3.87 a | 12.60 ± 0.89 a | 53.85 ± 2.95 a | - | - |
| | ID | 8.22 ± 1.90 b | 32.80 ± 5.54 c | 10.80 ± 2.17 b | 33.55 ± 5.64 b | - | - |
| 49 DAT | CA | 11.66 ± 1.21 a | 71.20 ± 3.63 a | 18.00 ± 2.24 a | 55.43 ± 3.42 a | - | - |
| | HLS | 12.54 ± 1.03 a | 68.20 ± 8.32 a | 15.20 ± 2.17 b | 57.91 ± 3.51 a | - | - |
| | WOA | 12.02 ± 0.45 a | 73.40 ± 5.94 a | 18.00 ± 0.71 a | 54.85 ± 2.17 a | - | - |
| | ID | 9.74 ± 0.56 b | 53.40 ± 2.30 b | 12.60 ± 1.67 c | 27.51 ± 5.13 b | - | - |
| 84 DAT | CA | 12.98 ± 1.28 ab | 101.60 ± 11.04 a | 23.00 ± 2.35 a | 54.21 ± 4.12 a | 57.75 ± 6.44 ab | 2.76 ± 0.28 a |
| | HLS | 13.84 ± 1.23 a | 102.60 ± 13.45 a | 23.80 ± 1.92 a | 56.03 ± 4.30 a | 61.95 ± 6.73 a | 2.82 ± 0.27 a |
| | WOA | 11.98 ± 0.67 bc | 103.40 ± 4.10 a | 21.20 ± 1.48 a | 48.77 ± 5.91 a | 53.40 ± 4.38 b | 2.67 ± 0.08 a |
| | ID | 11.40 ± 1.18 c | 64.60 ± 9.42 b | 17.00 ± 3.46 b | 17.34 ± 16.92 b | 32.05 ± 3.09 c | 0.77 ± 0.15 b |

DAT = days after transplant. CA = citric acid. HLS = humic-like substances. WOA = without organic amendment. ID = Fe deprivation. Values are means ± standard deviations, $n = 5$. [†] Within a row, values not sharing a letter are significantly different ($p < 0.05$).

### 3.2. Enzymatic Activity

Regardless of treatment, there was higher $H^+$-ATPase and PEPC activity in leaf tissue, whereas the FRO activity was higher in root tissue. Except for $H^+$-ATPase activity at 84 DAT, there were significant differences ($p < 0.001$) among the plant tissues studied.

### 3.2.1. $H^+$-ATPase Activity

The highest $H^+$-ATPase activity was documented at 27 DAT in leaf tissue. In this sampling, the CA and HLS treatments resulted in 61% and 51% reductions, respectively, compared to ID. At 49 DAT, the activity increased in root tissue, whereas it was reduced in leaf tissue compared to the first sampling; higher activity was documented under CA. During the fruit harvest stage (84 DAT), the activity of this enzyme increased in the leaf tissue, with 46% more activity in CA than in ID (Table 3).

### 3.2.2. PEPC Activity

In leaf tissue at 27 DAT, 84% higher PEPC activity was found in WOA, and 22% lower PEPC activity was found in plants receiving the HLS treatment relative to ID. On the other hand, at 49 DAT, the HLS and CA treatments promoted 55% and 50% increases in PEPC activity, respectively. Finally, at 84 DAT, plants treated with HLS showed a 150% increase in PEPC activity compared to ID. In root tissue, at 27 DAT, there was a PEPC-activity reduction of 39% and 67% in the HLS and CA treatments, respectively, whereas at 84 DAT, the CA and WOA treatments showed the lowest PEPC activity compared to ID (Table 3).

### 3.2.3. FRO Activity

In leaf tissue at 27 DAT, FRO activity was reduced by 12.5% in all treatments relative to ID, whereas at 49 and 84 DAT, the activity increased by 35.8% and 56.70% in the WOA treatment. On the other hand, similar FRO activity was found between treatments in root tissue, except at 27 DAT, where the CA treatment minimally reduced FRO activity compared to ID (Table 3).

**Table 3.** Enzymatic activity in tomato tissue treated with CA and HLS.

| Tissue | Sampling | Treatment | H⁺-ATPase | PEPC | FRO |
|---|---|---|---|---|---|
| Leaf | 27 DAT | CA | $1.46 \pm 0.18$ b[†] | $482.3 \pm 64.0$ ab | $7.2 \pm 0.2$ b |
| | | HLS | $1.84 \pm 0.07$ ab | $267.1 \pm 65.7$ cd | $7.4 \pm 0.1$ b |
| | | WOA | $1.57 \pm 0.05$ ab | $633.6 \pm 154.2$ a | $7.2 \pm 0.3$ b |
| | | ID | $3.76 \pm 2.22$ a | $343.5 \pm 56.9$ bc | $9.6 \pm 0.3$ b |
| | 49 DAT | CA | $0.92 \pm 0.11$ b | $834.2 \pm 151.4$ a | $7.7 \pm 0.2$ b |
| | | HLS | $0.89 \pm 0.04$ b | $856.5 \pm 140.6$ a | $7.5 \pm 0.3$ b |
| | | WOA | $0.89 \pm 0.05$ b | $784.9 \pm 194.0$ a | $9.1 \pm 0.4$ b |
| | | ID | $0.65 \pm 0.05$ c | $552.7 \pm 113.0$ a | $6.7 \pm 0.3$ b |
| | 84 DAT | CA | $2.65 \pm 1.33$ a | $465.6 \pm 221.8$ b | $10.7 \pm 1.2$ b |
| | | HLS | $1.58 \pm 0.06$ a | $759.6 \pm 136.5$ a | $8.2 \pm 0.2$ bc |
| | | WOA | $1.25 \pm 0.13$ a | $141.0 \pm 27.0$ cd | $10.5 \pm 1.1$ b |
| | | ID | $1.81 \pm 0.63$ a | $304.0 \pm 99.5$ bc | $6.7 \pm 1.1$ c |
| Root | 27 DAT | CA | $0.016 \pm 0.0005$ b | $51.5 \pm 29.7$ e | $15.8 \pm 2.1$ a |
| | | HLS | $0.017 \pm 0.0007$ b | $95.2 \pm 43.1$ de | $17.2 \pm 1.5$ a |
| | | WOA | $0.017 \pm 0.0004$ b | $191.8 \pm 16.0$ cde | $17.3 \pm 2.4$ a |
| | | ID | $0.017 \pm 0.0009$ b | $156.9 \pm 48.3$ cde | $14.5 \pm 1.3$ a |
| | 49 DAT | CA | $1.35 \pm 0.05$ a | $26.2 \pm 21.5$ b | $22.0 \pm 1.2$ a |
| | | HLS | $1.28 \pm 0.02$ a | $11.6 \pm 7.0$ b | $23.2 \pm 0.7$ a |
| | | WOA | $1.26 \pm 0.13$ a | $16.6 \pm 12.2$ b | $22.2 \pm 2.9$ a |
| | | ID | $1.32 \pm 0.05$ a | $4.6 \pm 0.1$ b | $25.3 \pm 1.5$ a |
| | 84 DAT | CA | $1.50 \pm 0.05$ a | $4.5 \pm 0.0$ d | $21.6 \pm 0.9$ a |
| | | HLS | $1.27 \pm 0.11$ a | $205.6 \pm 21.1$ bcd | $20.1 \pm 1.7$ a |
| | | WOA | $1.29 \pm 0.03$ a | $11.8 \pm 7.6$ d | $21.4 \pm 1.1$ a |
| | | ID | $1.48 \pm 0.04$ a | $261.9 \pm 16.5$ bcd | $20.1 \pm 1.8$ a |

DAT = days after transplant. CA = citric acid. HLS = humic-like substances. WOA = without organic amendment. ID = Fe deprivation. H⁺-ATPase activity expressed as: $\mu$mol NADH min$^{-1}$ mg$^{-1}$ prot. PEPC activity expressed as: nmol NADH min$^{-1}$ mg$^{-1}$ prot. FRO activity expressed as: nmol Fe$^{2+}$ reduced (g FW tissue) $^{-1}$ h$^{-1}$. Values are means $\pm$ standard deviations, $n = 5$. [†] Within a row, values not sharing a letter are significantly different ($p < 0.05$).

### 3.3. Gene Expression

The expression of the genes evaluated in leaf tissue is shown in Figure 1. The partial or complete overexpression and/or repression of a gene can be determined by comparing the expression of that gene against a critical threshold [37]. This threshold (value can be one) corresponds to the absolute value of a calibrator gene [38]. Based on this calibration, *LeHA1* was repressed to the highest degree in CA at 27 ($p < 0.01$) and 84 DAT (0.9- and 1.0-fold change, respectively), followed by WOA (1.0-fold change at 84 DAT, Figure 1a), whereas *LePEPC1* showed elevated expression in HLS at 27 DAT (0.6-fold change) compared to ID. However, *LePEPC1* was repressed under CA treatment at 27 and 49 DAT (up to 0.8-fold change; $p < 0.01$), whereas it showed elevated expression ($p < 0.01$) at 84 DAT (up to 1.1-fold change, Figure 1b). In the case of *LeFRO1*, high expression was observed at 27 and 49 DAT under the HLS treatment (4.10- and 4.19-fold change), whereas the CA treatment resulted in high expression at 49 DAT (2.33-fold change) and repression ($p < 0.01$) at 84 DAT. This gene exhibited elevated expression with WOA at 27 DAT (Figure 1c).

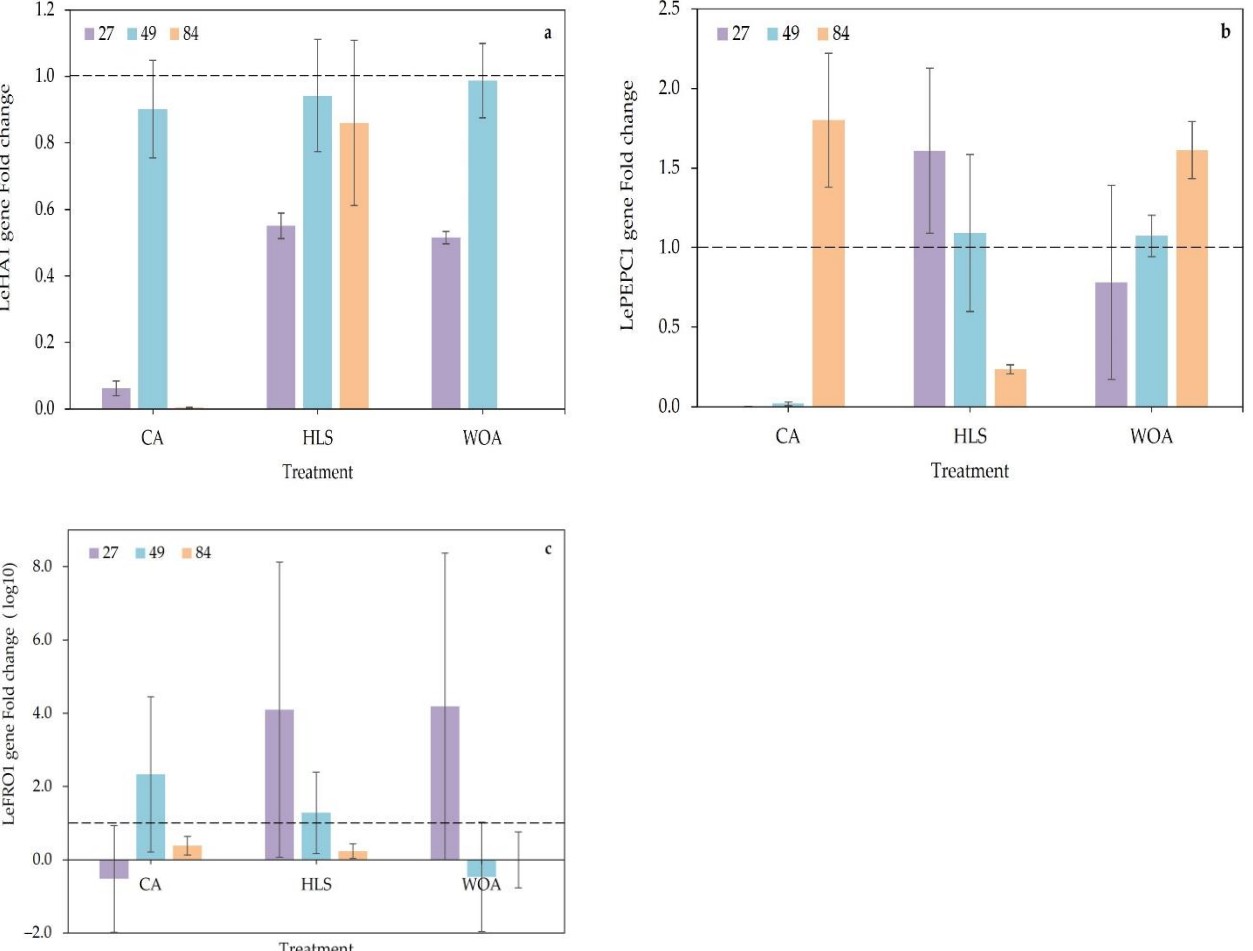

**Figure 1.** Expression of the *LeHA1* (**a**), *LePEPC1* (**b**), and *LeFRO1* (**c**) genes in tomato leaves of plants treated with CA and HLS. Values are means ± standard error (*n* = 5). The dashed line represents the constant value of the absolute control (ID).

Root gene expression was only evaluated at 49 and 84 DAT because increased expression of *FIT* (and probably *FRO1*, *HA1*, and *PEPC1*) has been associated with increased expression during the visible inflorescence stage (49 DAT) followed by a decrease during the fruit set-ripening stage (84 DAT) [18]. The behavior of gene expression in root tissue is presented in Figure 2. For *LeHA1*, elevated expression was documented at both sampling times; HLS at 49 DAT resulted in the greatest increase (3.8-fold), followed by WOA treatment (1.9-fold change), relative to ID. At 84 DAT, this gene showed 1.2- and 1.9-fold changes under HLS and WOA, respectively (Figure 2a). Regarding *LePEPC1* (Figure 2b), repression was documented to a greater extent at 84 DAT in CA (1.0-fold change) and HLS (0.8-fold change) relative to ID. *LeFRO1* exhibited repression; HLS application repressed (*p* < 0.01) *LeFRO1* up to 1.2- and 0.9-fold relative to ID at 49 and 84 DAT, respectively (Figure 2c).

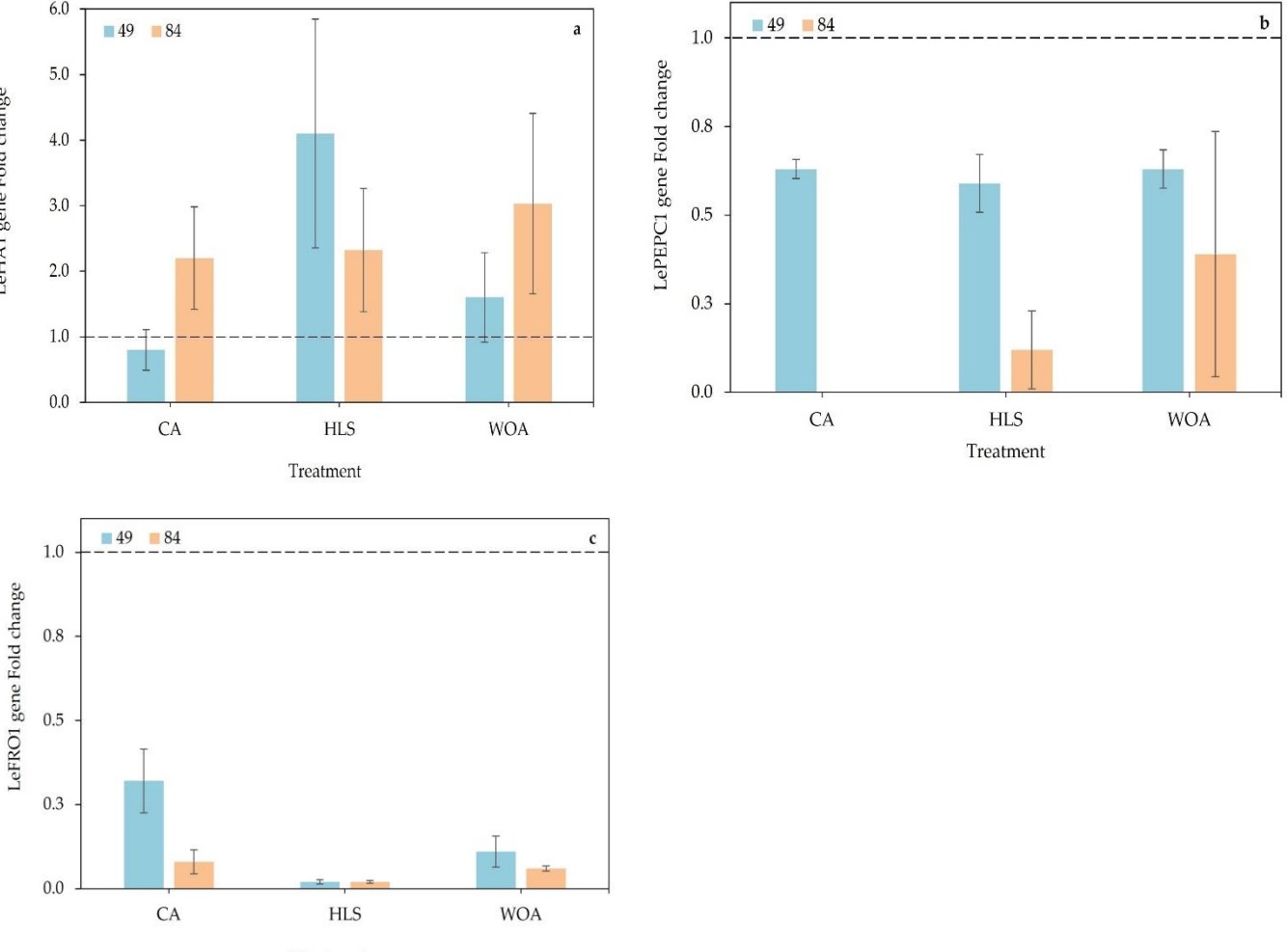

**Figure 2.** Expression of the *LeHA1* (**a**), *LePEPC1* (**b**), and *LeFRO1* (**c**) genes in tomato roots of plants treated with CA and HLS. Values are means ± standard error (*n* = 5). The dashed line represents the constant value of the absolute control (ID).

## 4. Discussion

Previous studies have found that the application of CA [39] and leonardite-derived compounds [21] improved the growth and yield of tomatoes grown in calcareous soil, probably via plant stimulation [40,41] and punctually by increasing photosynthetic parameters [42]. Similarly, the use of humic-Fe materials in soybean may be a suitable strategy for the management of Fe nutrition in calcareous soil [43], consequently leading to a reduction in Fe deficiency and an improvement in plant growth, development, and productivity. This is because the addition of Fe together with humic complexes act as "organic chelators" that boost plant growth [41]. CA and leonardite-derived complexes exhibit various levels of complexation with Fe, promoting plant growth by inducing a physiological response [44]. In addition, CA can reduce abiotic stress resulting from phosphorus deficiency in plants grown in calcareous soils [45–47]. Likewise, it has been described that the use of humic substances in calcareous soils promotes nutrient bioavailability by generating soluble complexes with minerals [23,48] and increasing nutrient uptake [40,41], and that their combination with chelate Fe (Fe-EDDHA) can increase the yield and nutrient content of plants [49].

Abiotic stress from Fe deprivation is signaled through the phloem from sinks in the root system, where there is an alteration in the absorption and transport of this nutrient [50]. In nongrass monocotyledons and in dicotyledons, such as tomato, that develop under conditions of Fe deprivation and/or in calcareous soils, strategy I presents an enzymatic

mechanism for coping with Fe-deficient conditions; according to the results obtained, $H^+$-ATPase and FRO enzymes may be involved in this mechanism and act together with enzymes of the antioxidant system in addition to the regulation of iron transporters (IRT1, ZIP and NRAMP1) and probably coumarin excretion [13]. Although our study does not show a situation of iron deficiency recovery, we can clarify the individual impact of CA and HLS on the enzymatic mechanisms related to iron metabolism in tomato plants grown in calcareous soil.

The $H^+$-ATPase enzyme generates electrochemical potential gradients by energizing ion channels and transporter proteins in the plasma membrane via the extrusion of $H^+$, with the consequent consumption of ATP [51]. Partial or total nutrient (e.g., Fe) deprivation in the growth medium, high carbonate–bicarbonate concentration, and/or high soil pH can induce an increase in root $H^+$-ATPase activity, acidifying the rhizosphere, solubilizing Fe, and preventing Fe precipitation [52]. Our results show higher activity of this enzyme in leaf tissue, which is probably due to its physiological role in the regulation of intra- and extracellular pH, loading of assimilates in the phloem, and redistribution of nutrients [51].

$H^+$-ATPase activity can be stimulated by humic substances [53]. Applications of humic acid purified from leonardite increased plasma membrane $H^+$-ATPase activity in cucumber roots and shoots [54,55]. This contrasts with what was found in the present study, where a reduction in tomato root tissue $H^+$-ATPase activity via HLS was documented; the reduction was probably derived from the promotion of soluble complexes that allowed Fe to exist in a bioavailable form in the soil [23,48] or via the acidifying effect of HLS [56]. This may also have been due to feedback control mechanisms and the concentration of $NO_3^-$ in the medium [55]. In contrast, the increase in the activity of this enzyme in leaf tissue (at 49 DAT) could be a consequence of the biostimulation (promoting a redistribution of nutrients) induced by HLS and CA [40,41]. In addition, CA is associated with the energy metabolism of the plant [44]; thus, $H^+$-ATPase activity would be necessary to generate gradients in the different cellular compartments of the leaf. The increase in root $H^+$-ATPase activity at 84 DAT under CA application implies an alteration in the gradients of the root system, as in radical cells, this enzyme plays an active role in xylem loading [51].

Tomato plants are thought to implement both strategies I and II; thus, in addition to extruding protons, reducing Fe, and promoting its transport, tomato plants also induce the synthesis of organic compounds (e.g., citrate, malate, and phenols, among others) [57]. In particular, CA tends to accumulate in the root under moderate conditions of Fe deprivation [58,59], where it acts as a reserve for chelating Fe and can present absorption windows over time [13]. The PEPC enzyme, which catalyzes the fixation of bicarbonate (in the presence of $Mg^{2+}$) to phosphoenolpyruvate, generating oxaloacetate and releasing Pi in C4 plants, could play a crucial role in this context. In nonphotosynthetic tissues and in C3 plants, PEPC enables anaplerotic reactions that provide intermediates in the Krebs cycle [30,58,60]. According to the pH-stat theory, PEPC would have to be forcibly activated to balance the pH change generated by the $H^+$-ATPase activity [30]; in this sense, the addition of acidifying compounds (such as CA and HLS) may impact PEPC activity via the contribution of carbonaceous skeletons.

According to previous studies, PEPC activity increases in roots under Fe-deficient conditions [30,58]. However, severe conditions of Fe deprivation (without Fe in nutrient solution) cause a reduction in the PEPC activity of the root system [59]. However, in this study, it was observed that the addition of HLS (+Fe chelated) promoted an increase in this enzyme (49 and 84 DAT), whereas CA and WOA reduced its activity (27 and 84 DAT). In the case of leaf tissue, the addition of CA and HLS increased PEPC activity. The modification of PEPC activity in the root and leaf tissue of tomato plants in response to CA and HLS application may be due to an alteration of anaplerotic reactions via the contribution of exogenous carbon skeletons [44], via a reprogramming mechanism in carbon metabolism under inadequate Fe conditions that facilitate root exudation [58], or via the acidic environment that can promote its exogenous addition [56].

The intracellular enzyme FRO is associated with the cell membrane and is responsible for reducing $Fe^{3+}$ to soluble, bioavailable $Fe^{2+}$ [61]. In leaf tissue, FRO activity is highly dependent on the Fe content in the cytoplasmic solution [62]. In this case (leaf tissue), a reduction in FRO activity was found at 27 DAT (in all treatments), which may imply the contribution and presence of Fe in its assimilable state ($Fe^{2+}$), whereas the increase in activity (49 and 84 DAT) may have been because Fe was present as $Fe^{3+}$ in the apoplast because of the content of $HCO_3^-$ and pH [60]; however, high bicarbonate content and pH apoplastic in plants may decrease FRO activity, causing Fe deficiency symptoms [63]. An increase in FRO enzyme activity has been documented in the roots of pea and tomato plants under Fe-deficient conditions [14,15,64], as was found in this work (27 and 84 DAT), which may suggest that the iron applied via nutrient solution is in its assimilable state ($Fe^{2+}$) via complexation with HLS or CA. The increase in the activity of this enzyme may or may not be accompanied by a greater number of secondary roots, which would imply a greater number of reduction zones [14].

Humic substances can reduce $Fe^{3+}$ to soluble forms due to their photocatalytic properties or their redox activity [65–67], contributing Fe to the chemical chelate and to the plant (as Fe-complex), which results in an increased concentration of soluble Fe in the soil and an enhanced translocation of Fe from the roots to the leaves [26]. However, the high $HCO_3^-$ content could precipitate $Fe^{2+}$ to $Fe^{3+}$ again, raising its concentration in the rhizosphere and stimulating root FRO activity, as seen in our results (49 DAT). A humic soil environment can condition FRO activity in the root plant, which could partially explain the behavior of this enzyme under our HLS treatment. A study carried out on cucumber reported that the application of humic acid purified to leonardite enhanced FRO activity at 72 h [54]. Repeated application of leonardite Fe humates can precipitate in the root, blocking entry pores in the cell wall and reducing Fe transport [68] as well as probably reducing FRO activity. However, a previous study demonstrated that the repeated application of 400 µL L$^{-1}$ of HLS on tomato plants potted in calcareous soils had a positive effect on Fe uptake [21]. Therefore, the reduction in the activity of this enzyme could not respond in the aforementioned sense. Despite this, caution should be exercised when applying humic substances. CA—a 6-carbon tricarboxylic compound [69]—is able to generate an acidic microenvironment in the rhizospheric zone, keeping Fe in its soluble and plant-available form and resulting in the low FRO activity (at least at 49 DAT) observed in our study.

The critical threshold of the calibrator gene values [38] allows us to determine the level of overexpression and/or total or partial repression of a gene [37]. In this context, *LeHA1*, *LeFRO1*, and probably *LePEPC1* may act as indicators of metabolically active Fe in the plant and in the growth medium [70].

Considering that the most suitable pH for the development of tomato plants is 6.0 [50], plants developed in calcareous soils with a high pH (8.5) and high $HCO_3^-$ content present a nutrient deficit response [13], triggering changes in the expression of specific genes [71]. Previous studies have documented increased expression of the H$^+$-ATPase enzyme promoter gene in Fe-deprived tomato, pea, and cucumber plants [15,16,25,72]. The data obtained here suggest that the modification of this gene can be assumed, in addition to nutrient input, to be a response to the soil pH and rhizospheric $HCO_3^-$ content because via these treatments, Fe (3 mg L$^{-1}$) was made available to the plant. In that sense, the repression of *LeHA1* (at 27 and 84 DAT) in leaf tissue under CA treatment and its high expression in root tissue under HLS treatment (at 49 DAT) suggest spatiotemporal expression patterns [14]. Such alterations would imply a modification of H$^+$ extrusion from the root tissue into the rhizospheric medium or cellular compartments [17,52]. In addition, humic complexes can induce an acid reaction in the soil by way of their diversity of functional groups [56], affecting *LeHA1* expression.

The application of purified humic acid from leonardite (2–250 mg L$^{-1}$ C) on cucumber grown under Fe-deficient conditions led to higher expression of the *CsHA2* gene (up to 4-fold over the control) and low expression of *CsHA1* (downregulated up to 5-fold) in the apical root [54]. These results show similar trends to ours; however, the present

work showed higher *LeHA1* expression in tomato roots without Fe deficiency and with continuous application of HLS. This discrepancy may be due to the differential response of the gene isoform [54,55], which leads to transient changes in H$^+$-ATPase activity.

Similarly, the gene encoding the PEPC protein showed increased expression in the root system under Fe-deficient environments [30,58]. Here, we found that *LePEPC1* was repressed in the roots of plants treated with CA and HLS with an Fe supply (in the form of EDTA) (Figure 2b). This suggests a reduced requirement for anaplerotic reactions due to the Fe supply in the growth medium. In the case of elevated leaf expression under HLS at 27 DAT, this may be indicative of active Fe in the plant [70].

In tomatoes, the *PEPC* gene family may be associated with other types of stresses, such as salinity and cold [73], but its expression could also be a means of coping with nutrient deficits (e.g., Fe) [59]. Applications of a mixture of organic acids (succinic, citric, malic, and oxalic acids, 100 μM) on alfalfa were reported to regulate the expression of the *PEPC* gene in roots under conditions of abiotic stress via Al [74].

The expression of the gene encoding FRO is strongly influenced, in calcareous soils, by the Fe content in the rhizospheric solution, the exposure time, and the HCO$_3^-$ content [14,75]. In plants, there is a higher expression of this gene in roots, shoots, and reproductive organs, whereas it is constitutively expressed in leaves [76]. In both cases, variations in the amino acid residues of *FRO1* are vital to maintaining the stability and high activity of the FRO enzyme [61].

Tomato and pea plants deprived of Fe show increased expression of *FRO1* in the root [15,64]. In this regard, several studies have shown that the application of leonardite humic acid in cucumber plants generates a higher expression of *CsFRO1* (between two- and ninefold) after the first days of application. Likewise, leonardite-derived substances applied to tomato plants subjected to Fe deficiency increase *LeFRO1* gene expression [70,77]. The use of an Fe complex with water-extractable humic substances (Fe-WEHS) or with citrate (Fe-citrate) generates a higher expression of *FRO*, *IRT1*, and *NRAMP* in cucumber [24], whereas in tomatoes under Fe deficiency, these compounds cause higher expression of *LeFRO1*, *LIRT1*, *LIRT2*, and *Ferritin2* in leaves [70,77].

In our study, we found repression of *LeFRO1* at 49 and 84 DAT (Figure 2c) in tomato roots with HLS application. This suggests that HLS enhances Fe bioavailability in addition to procuring soluble complexes in calcareous soil [23,48]. In the case of CA, its endogenous variations in tomato plants could be involved in the regulation of nuclear target gene expression [72]. In addition, the alteration in CA concentration affects the regulation of the tricarboxylic cycle [78]. In this sense, when applied exogenously, CA can act as a potential substrate of several metabolic pathways protecting the plant against abiotic stresses [69].

The discrepancy in transcriptional and post-transcriptional stimulation found in the present work may be due to the presence of a series of specific regulatory mechanisms for each level [54] or to associated feedback mechanisms prior to enzymatic activation or gene expression that increase Fe root concentration and Fe translocation [54]. However, the use of CA and/or HLS as an Fe chelate partner could supply stable Fe to the root, stabilize the chelated Fe, or generate an environment conducive to Fe absorption [70], thereby improving the growth and development of plants grown under calcareous soils. These mechanisms could reduce the power of the experimental design without affecting its validity, being necessary to establish unrestricted conditions to determine the precise effect on the supply of stable Fe of CA and HLS.

The enzymatic and gene expression responses observed in tomato plants (grown under calcareous soil) treated with CA and HLS offer a promising outlook in the prevention of Fe deficiency; however, as both compounds show complexation with Fe and stimulate the plant [44], it is necessary to verify possible synergistic effects when applied in combination.

## 5. Conclusions

Iron deprivation in tomato plants caused a reduction in growth, development, and yield. The application of CA or HLS (continuously or weekly, respectively) throughout

the crop cycle through a fertilizer solution improved the SPAD units, number of fruits harvested, and yield per plant. These substances also induced temporary alterations in enzyme activity ($p < 0.05$), reducing PEPC and FRO activity in roots. In leaf tissue, the CA treatment resulted in lower expression of *LeHA1* and *LePEPC1* ($p < 0.01$) as well as induced overexpression of *LeFRO1* ($p < 0.01$). In root tissue, HLS treatment resulted in *LeHA1* overexpression and *LePEPC1* and *LeFRO1* repression ($p < 0.01$), whereas CA repressed *LePEPC1* expression ($p < 0.01$). Thus, the use of CA and HLS may be a potential strategy in the management of ferric nutrition in tomato plants, as it can cause temporal alterations in enzyme activity (PEPC and FRO) and gene expression (*LeHA1*, *LePEPC1*, and *LeFRO1*) associated with iron uptake.

**Author Contributions:** Conceptualization, S.G.-M. and S.S.-G.; methodology, F.P.-L.; software, F.P.-L.; validation, A.B.-M. and A.J.-M.; formal analysis, F.P.-L.; writing—original draft preparation, F.P.-L.; writing—review and editing, F.P.-L. and S.G.-M.; funding acquisition, S.S.-G. All authors have read and agreed to the published version of the manuscript.

**Funding:** This research received no external funding.

**Data Availability Statement:** Not applicable.

**Acknowledgments:** The authors thank CONACyT and Arysta LifeScience for financial support.

**Conflicts of Interest:** The authors declare no conflict of interest.

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
