# Peer review of "Effects of Citric Acid and Humic-like Substances on Yield, Enzyme Activities, and Expression of Genes Involved in Iron Uptake in Tomato Plants"

_horticulturae, doi:10.3390/horticulturae9060630_

Round 1
Reviewer 1 Report
The topic discussed in this manuscript is very up-to-date, as the use of natural substances in crop production is a crucial issue today. This is important, among other things, because many active substances that can be important for successful crop protection are nowadays being withdrawn from use. Citric acid and humic acid and their derivatives occur in many cases in nature and are therefore relatively easily available. The nutrient uptake of tomatoes is particularly important as they are a very fast growing species and therefore nutrient-demanding.
The work carried out is complex, methodologically diverse and well supported.
Some issues to correct and supplement:
34 line "as a cofactor of antioxidant enzymes" what enzymes? list them!
Material and method
Lines 136-152 for the enzyme activity assays, please describe in more detail, as it is not clear why the 2x centrifugation was used, although an important part of this extraction is the recovery of the microsomal fraction. Not briefly, but in detail.
154 line RNA extraction: more detail please, as it was not extraction kit based, so the protocol used:starting material, key buffers and steps is needed, if of course not the fully detailed protocol is described.
How was the fold increase calculated for the gene expression studies? Which method was used?
205-208 The significant difference for all treatment combinations is an important result, but a more detailed explanation of the results is also needed at this point.
289-25920 "it achieves this mainly by enhancing H-ATPase and FRO activity" How did you conclude this? According to these results? Are the Authors sre that there are NO other enzymes that may be involved, since only these two were examined.
This is generalisation, should be more specific and inditcate, whther it is the owm results of the Authors, or this conclusion is based on literature data.
Author Response
We would like to thank the Editor and the reviewer for their thoughtful and positive comments on our manuscript. Detailed responses to each specific comment are provided in attached document.

Reviewer 2 Report
The manuscript entitled “Effects of Citric Acid and Humic-Like Substances on Yield, Enzyme Activities, and Expression of Genes Involved in Iron Uptake in Tomato Plants” investigated the effect of citric acid (CA) and humic-like compounds (HLS) on yield, enzymes, and genes involved in iron uptake in tomato.
The manuscript is quite well prepared. The significance and academic implications are potential. However, to be accepted by Horticulturae, authors should consider the following issues:
· The experimental design is simple and meager. The treatments are performed individually, not showing the actual recovery of the tomato plants from Fe deficient condition (calcareous soil?) after being applied by the CA and HLS. A combination of the two compounds should be tested.
· An explanation for the selection strategies of applied concentration of CA (0.1 mM) and HLS (400 µL/L) is requested.
· Abstract must be revised in order to clarify the link between calcareous soil and Fe deficiency. Additionally, the final statement in abstract and conclusion should be revised because the results did not show enzymatic modifications and gene alteration adequately.
· Sources of tomato, citric acid, and HLS must be provided. The detailed compositions of HLS are required.
· Discussion: The authors should not go too far into explaining the mechanism or reviewing previous studies. Instead, focus on highlighting the results of this study.
· Performance and styles should be revised according to the standard of the journal.
English should extensively be revised.
Author Response

(The authors gave the same response as above.)

Reviewer 3 Report
Iron deficiency is a common abiotic stress on plants growing in calcareous soils. Enzymatic activity is a mechanism for plants to access soil nutrients; enzymes such as H+-ATPase, PHOSPHOENOLPYRUVATE CARBOXYLASE (PEPC), and the intracellular enzyme FERRIC REDUCTION OXIDASE (FRO) are involved in Fe absorption. The effects of the application of citric acid (CA) and humic-like substances (HLS) on the yield, H+-ATPase, PEPC, and FRO enzyme activity, and expression of LeHA1, LePEPC1, and LeFRO1 genes in tomato plants grown under calcareous soil were studied. The results showed thatCA and HLS improved the SPAD units and increased the number of harvested fruits and yield per plant. Temporary alterations of enzyme activity were documented, which reduced PEPC and FRO activity in roots. In leaf tissue, CA resulted in lower expression of LeHA1 and LePEPC1, but induced LeFRO1 expression, while HLS application resulted in higher expression of LePEPC1 and LeFRO1. In roots, LeHA1 expression increased with HLS, while LePEPC1 and LeFRO1 showed lower expression with CA and HLS, respectively. Together,the application of CA and HLS through a nutrient solution in combination with Fe- chelate can improve Fe nutrition in tomato plants potted on calcareous soil through modifications at the enzymatic level and alterations in specific genes expression. In general, this paper is well-written and results are significant and I have only the following minor comments:
Miner comments:
1. For the two treatments, CA (0.1 mM + Fe gluconate-EDTA) and HLS (400 μL L-1 + 110 Fe gluconate-EDTA), why did you choose these two concentrations, any explanation for them?
2. Table 2, except the control ID, the other three treatments have no significant different from most the plant growth and harvesting parameters, author should also compare the different treatment in the result part;
3. For the sampling dates, why 27, 47, 89 days were chosen for sampling, author also mention it in the text;
4. Figure 1C and Figure 3B, the error bar even bigger than the value, is that result reliable, you should have more replicated so that you could reduce the error bar ?
5. Literature need to be updated, there is no literature from 2023.
Author Response

(The authors gave the same response as above.)

Round 2
Reviewer 1 Report
Dear Auhtors,
the concerns are addressed, only minor changes are required.
Please insert reference for the standard relative curve method (line 218).
Author Response
We have added the reference corresponding to the standard relative curve method:
- Larionov, A.; Krause, A.; Miller, W. A standard curve based method for relative real time PCR data processing. BMC Bioinformatics 2005, 6, 62, doi:10.1186/1471-2105-6-62.

Reviewer 2 Report
The authors addressed most of my comments.
However, the following point should be added to the discussion section
The experimental design is simple and meager. The treatments are performed individually, not showing the actual recovery of the tomato plants from Fe deficient condition (calcareous soil?) after being applied by the CA and HLS. A combination of the two compounds should be tested.
Future works will verify the combinatorial effect of treatments (CA+HLS) on iron deficiency recovery.
Author Response
We are grateful for the reviewer's recommendation, and we have argued the suggested point in the Discussion section:
… Although our study does not show a situation of iron deficiency recovery, we can clarify the individual impact of CA and HLS on the enzymatic mechanisms related to iron metabolism in tomato plants grown in calcareous soil.
… The above mechanisms could reduce the power of the experimental design without affecting its validity, being necessary to establish unrestricted conditions to determine the precise effect on the contribution of stable Fe by CA and/or HLS.
… The enzymatic and gene expression responses observed in tomato plants (grown under calcareous soil) treated with CA and HLS offer a promising outlook in the prevention of Fe deficiency; however, since both compounds show complexation with Fe and stimulate the plant [44], it is necessary to verify the possible synergistic effect when applied in combination.
The order of citations and references was also reviewed.
